# Sales Prediction by Integrating the Heat and Sentiments of Product Dimensions

**Xiaozhong Lyu [1],\* , Cuiqing Jiang [1,2,\*], Yong Ding [1,2], Zhao Wang [1] and Yao Liu [1]**

[1]   School of Management, Hefei University of Technology, Hefei 230009, China; dingyong@hfut.edu.cn (Y.D.);
     xcwangzhao@163.com (Z.W.); liuyaoemail@foxmail.com (Y.L.)

[2]   Key Laboratory of Process Optimization and Intelligent Decision Making of Ministry of Education,
     Hefei 230009, China

\*   Correspondence: adolflv@mail.hfut.edu.cn (X.L.); jiangcuiq2017@163.com (C.J.)

**Abstract:** Online word-of-mouth (eWOM) disseminated on social media contains a considerable amount of important information that can predict sales. However, the accuracy of sales prediction models using big data on eWOM is still unsatisfactory. We argue that eWOM contains the heat and sentiments of product dimensions, which can improve the accuracy of prediction models based on multiattribute attitude theory. In this paper, we propose a dynamic topic analysis (DTA) framework to extract the heat and sentiments of product dimensions from big data on eWOM. Ultimately, we propose an autoregressive heat-sentiment (ARHS) model that integrates the heat and sentiments of dimensions into the benchmark predictive model to forecast daily sales. We conduct an empirical study of the movie industry and confirm that the ARHS model is better than other models in predicting movie box-office revenues. The robustness check with regard to predicting opening-week revenues based on a back-propagation neural network also suggests that the heat and sentiments of dimensions can improve the accuracy of sales predictions when the machine-learning method is used.

**Keywords:** big data; sales prediction; online word-of-mouth; dynamic topic model; product attributes; back-propagation neural network

## 1. Introduction

Social media are forms of electronic communication (such as Facebook, WeChat, and IMDb.com) through which people create online communities to share information, ideas, personal messages, etc. Consumers increasingly use online word-of-mouth (eWOM) on social media for decision support before making purchases [1]. Therefore, social media marketing has recently appeared as an interdisciplinary and cross-functional concept that uses social media to achieve organizational goals by creating value for stakeholders [2]. Sales prediction is a foundation for social media marketing. Highly accurate and timely sales predictions can allow firms to reduce their profit losses and to improve their market performance [3]. Due to the superiority of big data on social media, sales predictions are being produced more than ever before to increase their accuracy and to enable them to support real-time marketing strategies for online retailers and enterprises. However, the accuracy of these models is still unsatisfactory. We need to extract more predictive information from high-frequency social media data to improve the accuracy of sales predictions.

High-frequency big data, such as eWOM on social media [4] and Google search index (GSI) data on the Google search engine [5], contain timely information and can improve the accuracy of sales predictions [6]. However, the accuracy of sales predictions is still unsatisfactory for irregular or nonseasonal sales trends [7,8]. Based on multiattribute attitude theory [9,10], we argue that the

heat and sentiments of product dimensions discussed in eWOM, which previous predictive models do not consider, can improve the accuracy of sales predictions. These factors have effects on product sales [11,12]. Therefore, this paper proposes a framework to simultaneously extract the heat and sentiments of product dimensions from eWOM and to then integrate them into a sales prediction model.

We chose the movie industry as our research context. We obtain reviews from IMDb.com, online search data from Google.com, and film-related data from BoxOfficeMojo.com. Finally, we construct a large dataset including data on films, Google Trends, and 349,269 reviews of 122 movies.

To extract the heat and sentiments of product dimensions, in this study, we developed a dynamic topic analysis (DTA) framework that integrates machine-learning techniques and lexicon-based methods. The framework has two major functions. First, DTA captures key product dimensions from eWOM without manual annotation. Second, DTA simultaneously extracts the heat and sentiments of the extracted dimensions. Next, we integrate the heat and sentiments of the dimensions to construct a new sales prediction model, called the autoregressive heat-sentiment (ARHS) model, to dynamically predict sales. We focused on the three most important dimensions discussed in movie eWOM: the *star*, the *genre*, and the *plot*. We found that the proposed ARHS model has better accuracy than previous models in predicting movie box-office revenues. Furthermore, the ARHS model can predict sales of all kinds of products if the products have multiple attributes and sufficient eWOM. The robustness check with regard to forecasting opening-week revenues using a back-propagation (BP) neural network demonstrates that the predictive model integrating the heat and sentiments of dimensions is more accurate.

## 2. Literature Review

eWOM influences consumer purchase intentions by changing the preferences for alternatives and in turn influences product sales based on information theory [13,14]. We introduce multiattribute attitude theory in this research domain.

### 2.1. eWOM's Effect on Sales

Some research on eWOM has shown mixed findings regarding the direct effects of eWOM on product sales [15,16]. Other research shows the moderating effects of rating variance [17], review helpfulness [18], and the features of reviewers [19], products [20,21], and social media platforms [22–24]. In this paper, we focus on the direct effects of eWOM.

The volume of eWOM represents the popularity (overall heat) of products supplied by reviewers, such as the number of online reviews. Previous studies have found mixed results regarding the effects of eWOM volume on sales [25,26]. Many studies have found that eWOM volume positively affects sales [25,27–29], whereas several other studies have not found a significant effect [30,31]. Xu [32] found that more information could even reduce sales under certain conditions. Therefore, under some conditions, volume cannot be used to predict sales. The multiattribute attitude model demonstrates that only the most important attributes that reflect consumers' perceptual dimensions can influence consumer purchasing decisions [33]. In this paper, we divide the overall heat of products into the heat of their important attributes. Previous research has proven that the heat of key product dimensions can influence product sales [11]. In this paper, we demonstrate that the heat of dimensions has predictive power in predicting movie box-office revenues.

The valence of eWOM can be the average rating on the rating scale (e.g., 1–5), or it can be binary (positive and negative). It can also be regarded as the overall sentiment of eWOM [34,35]. Most studies have reported a significant, positive effect of valence [27,36], but other studies have not found a significant effect [29,37]. The overall sentiment of eWOM represents the emotion conveyed by reviewers to consumers. However, the overall sentiment represents the aggregation of the sentiments of all attributes discussed in eWOM, which include irrelevant and abundant attributes. Perhaps for this reason, prior studies have found that the overall sentiment of eWOM has no effect on movie box-office revenues [38,39]. Chen and Xie [40] demonstrate that eWOM provides product-dimension preference

information that helps consumers find products that match their needs. Potential consumers will change their purchase intentions regarding a product after perceiving the sentiments of important product dimensions from online reviews [12]. We argue that analyzing the eWOM sentiments of key product dimensions can provide new insights for sales prediction and overcome the weakness of overall sentiment.

### 2.2. eWOM-Based and GSI-Based Sales Prediction

Online search data include indexes (from zero to 100) of the frequency of the object searched in an online search engine, such as Google.com. This type of data has been used to predict movie box-office revenues [41]. Bughin [42] finds that the valence of eWOM influences sales more than Google Trends. Geva et al. [5] find that adding Google search data to models based on the more commonly used eWOM data improves the accuracy of sale predictions for search products. Regarding the different natures of search products and experience products, the effect of online information is always different in these two kinds of products [27]. In this paper, we aim to demonstrate whether the model used for search products is valid for experience products. Geva et al. [5] also found that for search products, Google search index (GSI) models based on inexpensive Google Trends provide accuracy that is at least comparable to that of eWOM-based prediction models. These studies have proven that both online search data and eWOM have powerful predictive ability. To date, however, the predictive ability of the heat and sentiments of product dimensions has not been researched. This study attempts to improve the prediction accuracy of movie box-office revenues by proposing a comprehensive model that simultaneously integrates the heat and sentiments of product dimensions.

## 3. Materials and Methods

### 3.1. Research Framework

Figure 1 shows the framework of our study; it will help researchers develop a sales prediction model for products with abundant eWOM and multiple attributes. First, based on the findings that eWOM volume [25,27–29] and ratings [27,36] positively affect sales, we developed the eWOM model by integrating eWOM variables into the autoregressive (AR) model. Additionally, we developed the GSI model by integrating Google Trends into the AR model based on findings in the literature [41]. We then integrated Google Trends into the eWOM model following the method of [5] and named this benchmark model the autoregressive online (ARO) model.

The above models consider volume (heat) and valence as a whole. Multiattribute attitude theory decomposes a consumer's overall attitude toward a product into smaller components [9,10]. These components, which are the most important attributes, reflect only consumers' perceptual dimensions rather than product characteristics that are directly controllable and measurable by marketing managers [33]. Multiattribute attitude theory shows that only the importance and valence of important product attributes can predict consumer purchase predispositions. Volume and sentiment, which represent the sum of the heat and sentiments of all attributes implied in eWOM, may have little effect on sales because of the offset effects of irrelevant and redundant attributes. This may be why some research finds that the volume and valence of eWOM have no effect on sales [15,38,39]. In previous studies, attributes were generated based on expert judgment and in-depth interviews. Additionally, the measures of attribute importance and valence in the attitude model were obtained by surveying a sample of respondents [43]. Recent research shows that topic models can be used to generate important attributes that reflect consumers' perceptual dimensions, and the heat and sentiment of these product dimensions can be used as a proxy for attribute importance and valence [44–46]. Therefore, the heat and sentiments of dimensions are more suitable than eWOM volume and valence for predicting sales.

Thus, according to the discussion above, we hypothesize the following:

**H1**. *The heat and sentiments of dimensions have predictive power for sales and can improve the performance of the benchmark model.*

Finally, to verify the hypothesis, we used DTA to extract the heat and sentiments of product dimensions from eWOM and integrated them into the benchmark model to determine whether the new model, the ARHS model, has better prediction accuracy.

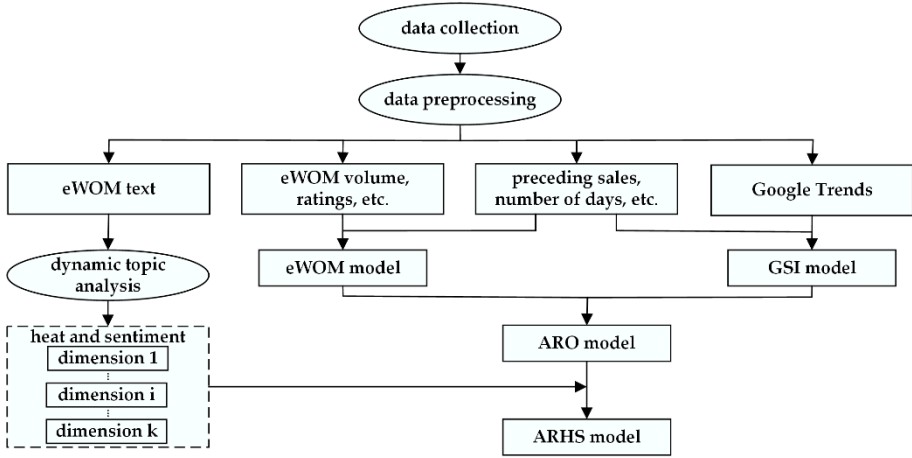

**Figure 1.** Framework for constructing the autoregressive heat-sentiment (ARHS) model.

*3.2. Data and Variables*

3.2.1. Data Collection

This paper focuses on online reviews of movies because within the film industry, online reviews are more popular than other types of eWOM [29]. In this study, IMDb.com, Google.com, and BoxOfficeMojo.com are the data sources. We examined movie reviews on IMDb.com, the most popular and authoritative information source for movies worldwide, for approximately seven weeks after movie releases. We then collected daily data on box-office revenues, budgets, and distributors as well as other movie information from BoxOfficeMojo.com. Our unit of time was one day; however, we aggregated the reviews published before the release day into one time window. The weekly Google Trends before a movie release and the daily Google Trends one day before and 49 days after a movie release were obtained and reviewed. Because the unit of time was one day, we had a sufficiently long study period with enough observations to provide credible results. The final dataset included Google Trends and eWOM information for 50 consecutive time windows and revenue information for 49 time windows. All movies were released in the US from 2010 to 2016.

After filtering out movies with fewer than 100 reviews by the end of the data period, we identified 349,269 reviews for 122 sample movies. We chose a threshold of 100 reviews to ensure that we have sufficient reviews to train the topic model used in the DTA. As shown in Table 1, our sample movies exhibited great diversity in terms of film distributor, movie genre, release month, and Motion Picture Association of America (MPAA) rating. Table 2 indicates that the total domestic gross and production budgets of the movies are right-skewed; that is, all but a few movies have low box-office revenues and product budgets.

Tables 3 and 4 list the definitions of and statistics regarding eWOM, Google Trends, and our film-related variables. First, we measured the eWOM volume and valence, which are represented by $v_{t,1}$ and $v_{t,2}$, respectively. Volume is the log-transformation of the daily number of reviews. We added one to the daily number of reviews to ensure that the log-transformation result was not negative [47]. Valence is the mean of the daily review ratings, reflecting the overall sentiment of reviewers with regard to a specific movie [28]. If there were no reviews on one day, then we used the average valence of the preceding days as a proxy [48]. Second, we used the variable $v_{t,3}$ to denote the number of days since the movie release to consider the time effect. Third, we set the dummy variable $v_{t,4}$ to one if the day was on the weekend and zero otherwise to consider the seasonal effect. Fourth, the variable $v_{t,5}$ represents the number of cinemas at which a film was being shown [17]. Finally, we used the

Google Trends of movie names, and the initial trends range from 0 to 100 in terms of online search data. Table 4 shows that sales, volume ($v_{t,1}$), and theaters ($v_{t,5}$) have right-skewed distributions and that the skewness of volume and sales is very large. This result means that very few movies had high box-office revenues or high heat and that most movies had low box-office revenues or low heat. The distributions of valence ($v_{t,2}$) are relatively evenly distributed.

**Table 1.** The diversity of movies.

| Distributor | Freq. | Genre | Freq. | Release Month | Freq. | MPAA Ratings | Freq. |
|---|---|---|---|---|---|---|---|
| Warner Bros. | 18 | Drama | 38 | January | 10 | R | 57 |
| Lionsgate | 16 | Comedy | 37 | February | 11 | PG-13 | 50 |
| Paramount | 12 | Thriller | 14 | March | 12 | PG | 14 |
| Weinstein | 10 | Action | 13 | April | 7 | NC-17 | 1 |
| Fox | 10 | Sci-Fi | 10 | May | 10 | Total | 122 |
| Sony | 9 | Horror | 9 | June | 6 | | |
| Universal | 7 | Animation | 8 | July | 7 | | |
| Open Road Films | 7 | Crime | 6 | August | 11 | | |
| Focus Features | 6 | Fantasy | 5 | September | 11 | | |
| Roadside Attractions | 6 | Adventure | 3 | October | 12 | | |
| FilmDistrict | 4 | Sports | 2 | November | 11 | | |
| Relativity | 4 | Music | 2 | December | 14 | | |
| Buena Vista | 4 | Romance | 2 | | | | |
| CBS Films | 2 | Documentary | 1 | | | | |
| Bleecker Street | 2 | War | 1 | | | | |
| TriStar | 2 | | | | | | |
| A24 | 1 | | | | | | |
| Radius-TWC | 1 | | | | | | |
| Rogue Pictures | 1 | | | | | | |

**Table 2.** The distribution of movie gross and budgets.

| Domestic Gross (Million) | Freq. | Production Budget (Million) | Freq. |
|---|---|---|---|
| $\leq 25$ | 40 | $\leq 25$ | 59 |
| 25–50 | 32 | 25–50 | 30 |
| 50–75 | 21 | 50–75 | 11 |
| 75–100 | 10 | 75–100 | 7 |
| 100–125 | 9 | 100–125 | 3 |
| 125–150 | 1 | 125–150 | 6 |
| 150–175 | 3 | 150–175 | 1 |
| 175–200 | 3 | 175–200 | 5 |
| 200–225 | 1 | Total | 122 |
| 225–250 | 1 | | |
| 250–275 | 1 | | |

**Table 3.** Key variables for each movie: numerical.

| Variable | Description (for Each Movie) | Measure and Data Sources |
|---|---|---|
| Sales | Daily domestic box-office revenues | Dollars (log-transformation); BoxOfficeMojo.com |
| $v_{t,1}$ | Daily number of reviews | Number (log-transformation); IMDb.com |
| $v_{t,2}$ | Daily valence of reviews | Average of daily ratings (0–10); IMDb.com |
| $v_{t,3}$ | Days from initial release | Number (1–49) |
| $v_{t,4}$ | Whether the day is on the weekend | 1 = the day is on the weekend (Fri, Sat, and Sun), 0 = others |
| $v_{t,5}$ | Daily number of cinemas | Number (log-transformation); BoxOfficeMojo.com |
| $v_{t,6}$ | Daily Google Trends of movie name | Number (0–100); Google.com |

**Table 4.** Summary statistics of the key variables.

| Variable | Mean | Median | Maximum | Minimum | Std. Dev. | Skewness | Kurtosis |
|---|---|---|---|---|---|---|---|
| Sales | 1,039,207 | 263,875 | 35,167,017 | 10 | 2,204,483.7 | 5.351 | 46.855 |
| $v_{t,1}$ | 22.11526 | 11 | 506 | 0 | 38.754737 | 3.956 | 26.541 |
| $v_{t,2}$ | 3.801627 | 4 | 10 | 0 | 3.6083277 | 0.162 | 1.410 |
| $v_{t,5}$ | 1483.412 | 1195 | 4324 | 1 | 1264.8718 | 0.343 | 1.634 |
| $v_{t,6}$ | 33.49281 | 28 | 100 | 2 | 21.922873 | 1.101 | 3.815 |

Figure 2a shows the relationship between Google Trends and the box-office revenues of the movie Gravity. Figure 2b shows the relationship between eWOM volume and the box-office revenues of the movie Gravity. We observe that both the eWOM and GSI data have high correlations with movie box-office revenues.

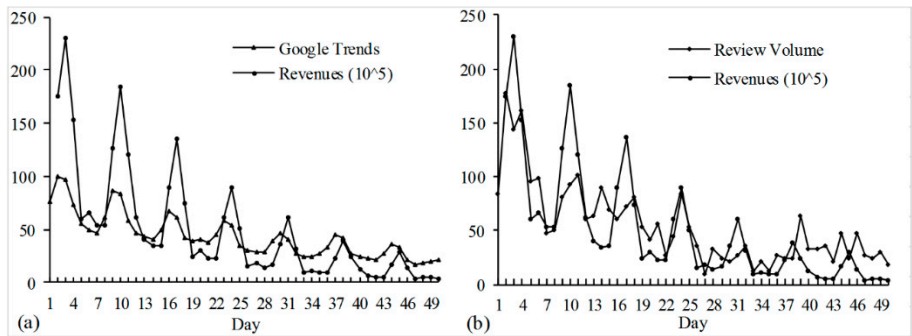

(a)  (b)

**Figure 2.** The relationship between online information and the box-office revenues of the movie Gravity. (**a**) The relationship between Google Trends and box-office revenues; (**b**) the relationship between the number of reviews and box-office revenues.

3.2.2. Dynamic Topic Analysis

For 122 movies, we constructed a DTA framework by integrating the dynamic topic model (DTM) [49], the lexicon-based method [50], and the Stanford natural language processing (NLP) technique [51] to derive the heat and sentiments of dimensions from online reviews. We obtained 122 daily documents by integrating hundreds of daily reviews for each movie into one document. Finally, the daily documents compiled over 50 days constitute our review corpus, which contains 349,269 reviews. Figure 3 shows the structure of the corpus.

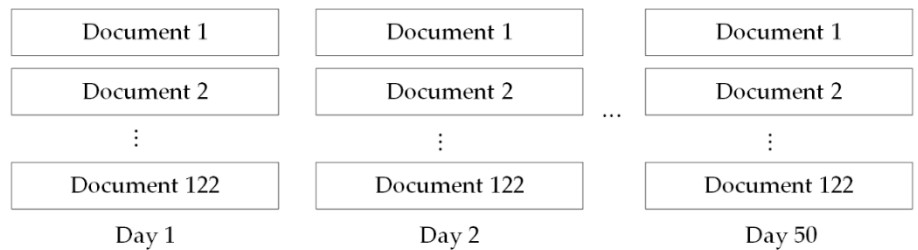

**Figure 3.** The structure of our review set.

We preprocessed each document by using the steps used in a study by Guo et al. [45]. First, we eliminated non-English words and spelling errors, such as web sites, punctuation marks, and numbers. We then used the Stanford NLP package for word text tokenization, part-of-speech tagging, and word stemming. Finally, each document became a word-of-bag.

To extract key product dimensions from a large corpus of text data in an effective manner, previous studies have used the latent Dirichlet allocation (LDA) model [44,45]. The DTM is more suitable for extracting key product dimensions from our structured review set [49] and is an extended

LDA method [52]. The DTM can quickly identify a conglomeration of connected topics from a very large number of documents over different time windows, which LDA alone cannot do.

As a machine-learning method, the DTM is highly efficient in handling online big data. We used the DTM to extract key product dimensions, the heat of these dimensions, words that represent each dimension and the changes in these factors over different time windows. The DTM assumes that a review comprises a sequence of $N$ words, $d = (w_1, w_2, \ldots, w_N)$, $D$ reviews form a review set, $C_t = [d_1, d_2, \ldots, d_D]$, and $T$ review sets form a corpus over $T$ time windows, $C = \{C_1, C_2, \ldots, C_T\}$. The DTM also assumes that reviewers share $K$ dimensions across the corpus over the $T$ time windows. In each time window, the DTM assumes that reviewers express their experience with a product or service over $K$ dimensions. For instance, a reviewer may comment about a movie in a review by focusing on three dimensions with different heat and sentiments: 30% and 4.9 for movie stars, 40% and 3.4 for the story plot, and 30% and 2.1 for the background music. Thirty percent is the dimension heat of movie stars, which means that one-third of the review is about movie stars; additionally, 4.9 is the sentiment strength of movie stars, which means that the reviewer has a strong sentiment toward movie stars.

Comparing the perplexity of the DTM and the semantics of the dimensions when using different values of $K$, we determine the optimal number of key product dimensions [11]. Ultimately, we find three movie dimensions that can perfectly represent the review corpus. The formula for the perplexity of the DTM for the document set on day $t$ is as follows:

$$perplexity(C_t) = exp\left(-\frac{\sum_{d=1}^{D}\sum_{n=1}^{N_d} \log \sum_{k=1}^{K} p(W_{d,n} = w|Z_{d,n} = k)p(Z_{d,n} = k|d)}{\sum_{d=1}^{D} N_d}\right) \tag{1}$$

where $C_t$ is the document set on day $t$; $D$ is the number of documents on day $t$; $N_d$ is the number of words in document $d$; $K$ is the number of dimensions; $p(W_{d,n} = w|Z_{d,n} = k)$ is the heat of word $w$ in dimension $k$; $p(Z_{d,n} = k|d)$ is the heat of dimension $k$ in document $d$. DTM learning with Gibbs sampling can simultaneously generate the heat of the words in each dimension and the heat of the dimensions in each document. Readers can refer to [49] for details on the DTM. Let $\vartheta_{i,t}$ be the heat of the $k^{\text{th}}$ dimension of the $i$th movie on day $t$. $\vartheta_{i,t}$ can be calculated as follows:

$$\vartheta_{i,t,k} = \frac{\sum_{d=1}^{D_{i,t}} p(Z = k|t, d, i)}{D_{i,t}} \tag{2}$$

where $p(Z = k|t, d, i)$ is the heat of dimension $k$ in document $d$ of movie $i$, and $D_{i,t}$ is the number of documents for movie $i$ on day $t$. In our research context, $D_{i,t}$ equals one.

We name the three dimensions *plot*, *star*, and *genre*, following the method of Guo et al. [45]; these dimensions have been regarded as the three most important attributes of movies [7,53]. Table 5 shows the changes in the dimension *plot* in different time windows.

**Table 5.** The change in words and the weight of the dimension *plot*.

| *plot* | weight | *plot* | weight | *plot* | weight |
|---|---|---|---|---|---|
| story | 0.9% | plot | 0.5% | plot | 0.5% |
| plot | 0.4% | story | 0.4% | story | 0.4% |
| book | 0.4% | book | 0.3% | book | 0.4% |
| horror | 0.3% | horror | 0.3% | horror | 0.3% |
| dark | 0.2% | dark | 0.3% | dark | 0.2% |
| original | 0.3% | original | 0.2% | original | 0.2% |
| scary | 0.2% | scary | 0.2% | scary | 0.2% |
| real | 0.2% | maze | 0.2% | maze | 0.2% |
| pretty | 0.2% | pretty | 0.2% | pretty | 0.2% |
| action | 0.2% | love | 0.2% | house | 0.2% |

The heat of a dimension refers to the proportion of reviewers' discussion concerned with the dimension of a product in eWOM. For example, the heat of the dimension *plot* denotes the proportion of consumers' discussion concerned with *plot*-related information in reviews. Figure 4 shows the changes in the heat of the three movie dimensions over 50 days. Consumers talk more about movie stars and the story plot in the early days after a movie's release than they do at the end of the release.

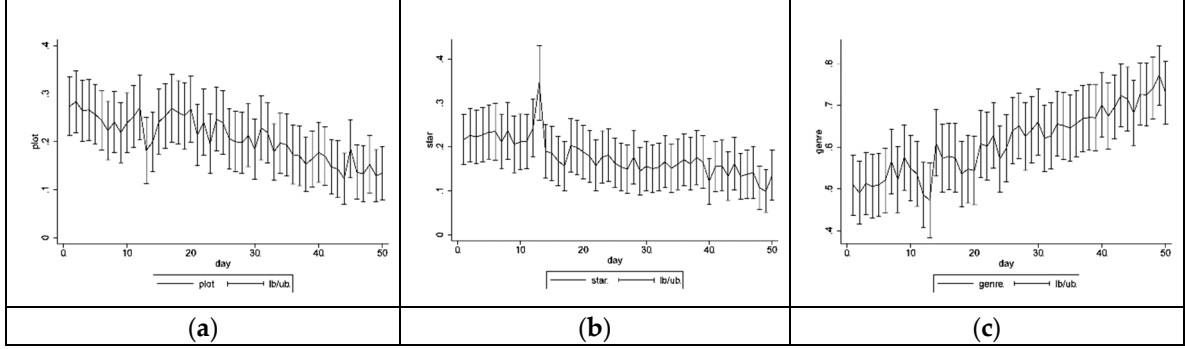

**Figure 4.** Average heat of the three dimensions for 122 movies. (**a**) The average heat of the dimension *plot*; (**b**) the average heat of dimension *star*; (**c**) the average heat of dimension *genre*.

We then used the sentiment lexicon and syntax relation to calculate the sentiments of dimensions. Lexicon-based methods that use a publicly recognized sentiment lexicon are more objective and suitable for big data sentiment analysis than machine-learning-based methods that require expert annotations because expert annotation has a high cost and there are artificial deviations. Most studies on dimension sentiment analysis divide dimensions into positive and negative classes [54], and sentiment analysis methods are different based on different applications. We calculated the sentiment strength of each dimension that can forecast movie box-office revenues. We extracted the syntactic relations between the dimension words and sentiment words in the daily review sentences using the Stanford NLP package, and we obtained the sentiments of the dimension words based on the extracted relations. Table 6 shows the main sentiment mining rules used in our framework.

**Table 6.** The main rules for mining the sentiments of dimension words.

| Syntax Relations | Examples | Word Sentiments |
| --- | --- | --- |
| Nominal subject | The *plot* is *boring*. | *Plot*: 3.0 |
| Adjectival modifier | She is a *good actor*. | *Actor*: 3.8612 |
| Direct object | I *enjoy 3D*. | *3D*: 3.9782 |
| Open clausal complement | I think the actor *enjoys acting*. | *Acting*: 3.9782 |
| Adverb modifier | Tom *performed earnestly*. | *Perform*: 3.5 |
| Relative clause modifier | I saw an *actor* who people *dislike*. | *Actor*: 3.5417 |

Finally, we calculated the average daily sentiment strength of the dimensions for each movie. Let $s_{i,n,d}$ be the sentiment value for the $n^{th}$ dimension word at the $i^{th}$ time (location) in document $d$ for one movie. The sentiment of the $k^{th}$ dimension for one movie on the $t^{th}$ day can then be formulated as follows:

$$\theta_{t,k} = \frac{1}{N} \sum_{n=1}^{N} \frac{1}{D} \sum_{d=1}^{D} \frac{1}{I} \sum_{i=1}^{I} s_{i,n,d}. \tag{3}$$

Intuitively, $\theta_{t,k}$ represents the average strength of the sentiment of the $k^{th}$ dimension. Figure 5 shows the average sentiments of the dimension *plot* for 122 movies. Using Figures 4 and 5, we can easily monitor consumer feedbacks (heat and sentiments) on product dimensions over time.

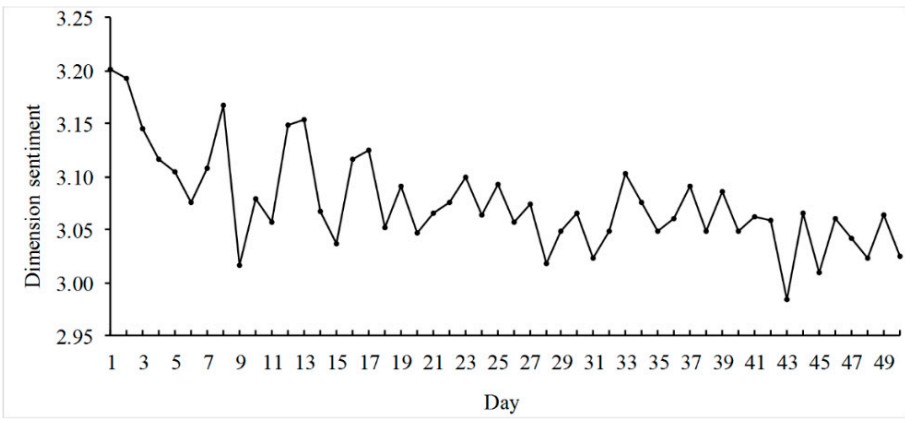

**Figure 5.** Average sentiments of the dimension *plot* for 122 movies.

In Table 7, we describe the key variables of the dimensions.

**Table 7.** Key variables for each movie: dimensions.

| Variable | Description | Measures |
|---|---|---|
| $\vartheta_{t,1}$ | The heat of the dimension *plot* on day $t$ | Probabilistic |
| $\vartheta_{t,2}$ | The heat of the dimension *star* on day $t$ | Probabilistic |
| $\vartheta_{t,3}$ | The heat of the dimension *genre* on day $t$ | Probabilistic |
| $\theta_{t,1}$ | The sentiment of the dimension *plot* on day $t$ | Numerical value |
| $\theta_{t,2}$ | The sentiment of the dimension *star* on day $t$ | Numerical value |
| $\theta_{t,3}$ | The sentiment of the dimension *genre* on day $t$ | Numerical value |

Table 8 shows the summary statistics of the variables. The heat of the dimensions ($\vartheta_{t,i}$) is between zero and one. The median of the sentiments of the dimensions ($\theta_{t,i}$) is three.

**Table 8.** Summary statistics of the dimension variables.

| Variable | Mean | Median | Maximum | Minimum | Std. Dev. | Skewness | Kurtosis |
|---|---|---|---|---|---|---|---|
| $\vartheta_{t,1}$ | 0.26932 | 0.00971 | 0.9999957 | $1.93 \times 10^{-6}$ | 0.4063216 | 1.082 | 2.284 |
| $\vartheta_{t,2}$ | 0.13574 | 0.00971 | 0.9999957 | $2.16 \times 10^{-6}$ | 0.3078812 | 2.184 | 5.992 |
| $\vartheta_{t,3}$ | 0.59495 | 0.95943 | 0.9999949 | $1.43 \times 10^{-6}$ | 0.4544953 | −0.426 | 1.250 |
| $\theta_{t,1}$ | 3.07341 | 3 | 4.83333 | 0.130435 | 0.3685301 | −3.347 | 28.987 |
| $\theta_{t,2}$ | 3.09445 | 3 | 4.90476 | 0.130435 | 0.3519009 | −2.837 | 28.437 |
| $\theta_{t,3}$ | 3.08610 | 3 | 4.60417 | 0.130435 | 0.299457 | −3.282 | 35.261 |

### 3.3. Predictive Model

We used the first 40 days of data to train the predictive model and the last 9 days of data to test the trained model. The regressive model can have better forecasting performance than the machine-learning models when the amount of relevant information is sufficient and when the variation in box-office revenues is small [55]. However, if there is not enough information, then the machine-learning model can help improve the forecasting accuracy by more thoroughly utilizing the limited information given. According to the sufficient predictors discussed in Section 3.2 and the relatively stable revenues in the test period, the proposed approach that we constructed was based on the autoregressive model because the regressive model is the most efficient predictive model [7]. We also needed to address some methodological concerns. First, we log-transformed some skewed variables to give them similar normal distributions. Second, we used the variance inflation factor (VIF) to assess multivariate multicollinearity. The VIF values were lower than the threshold of five; thus, multicollinearity was not a serious issue [56].

### 3.3.1. Autoregressive Model

We started with an AR model as our base model to forecast movie box-office revenues. We used this AR model with the parameter $p$ to model the relationship between preceding box-office revenues and current box-office revenues as follows:

$$\log(Sales_t) = \alpha + \sum_{i=1}^{p} \varphi_i \log(Sales_{t-i}) + \epsilon_t \tag{4}$$

where $\varphi_1, \varphi_2, \ldots, \varphi_p$, are the parameters to be estimated, $\alpha$ is the effect of the combination of time-invariant variables, such as the production budgets and genres of movies, and $\epsilon_t$ is an error term. The AR model uses only preceding sales to predict current or future sales.

### 3.3.2. ARO Model

In addition to preceding box-office revenues, online information, such as Google Trends and eWOM volume, might greatly influence box-office revenues. According to the discussion above, we propose a predictive model by integrating online information into the AR model. This model includes all the variables of previous GSI models and eWOM models. Our ARO model is similar to that proposed in [5], and it can be formulated as follows:

$$\log(Sales_t) = \alpha + \sum_{i=1}^{p} \varphi_i \log(Sales_{t-i}) + \sum_{i=0}^{q} \sum_{j=1}^{J} \rho_{i,j} v_{t-i,j} + \epsilon_t \tag{5}$$

where $v_{t,j}$ represents the $j^{th}$ online information variable on day $t$. We determined $p$ and $q$ by comparing model accuracy when using different values of $p$ and $q$. $\varphi_i$ and $\rho_{i,j}$ are parameters that need to be estimated. The parameter $q$ specifies the lags of the preceding days of the online information variables; $J$ indicates the number of these variables. The ARO model uses preceding sales, Google Trends, the eWOM variables and other predictors in Table 4 to predict current and future sales.

### 3.3.3. The ARHS Model

According to previous studies, the heat and sentiments of product dimensions are very important for sales [11,12]; thus, it is desirable to integrate the heat and sentiments of movie dimensions into predictive models to achieve better accuracy. In this section, we extend the ARO model to the ARHS model. We formulate the ARHS model as follows:

$$\log(Sales_t) = \alpha + \sum_{i=1}^{p} \varphi_i \log(Sales_{t-i}) + \sum_{i=0}^{q} \sum_{j=1}^{J} \rho_{i,j} v_{t-i,j} + \sum_{i=0}^{\gamma} \sum_{k=0}^{K} \omega_{i,k} \vartheta_{t-i,k} + \sum_{i=0}^{\delta} \sum_{k=0}^{K} \mu_{i,k} \theta_{t-i,k} + \epsilon_t \tag{6}$$

where $p$, $q$, $\gamma$, and $\delta$ are user-defined parameters, $\epsilon_t$ is an error term, and $\varphi_i$, $\rho_{i,j}$, $\omega_{i,k}$, and $\mu_{i,k}$ are parameters that need to be estimated. $\vartheta_{t,k}$ and $\theta_{t,k}$ are the heat and sentiments, respectively, of the $k^{th}$ dimension at time $t$, and are obtained by using DTA. $p$, $q$, $\gamma$, and $\delta$ specify how far the model "looks back" into the past, whereas $J$ and $K$ specify how many related variables we would like to consider. $J$ and $K$ are fitted as described in Section 3.1. We used the least squares method to train all the models. The ARHS model extends the ARO model by integrating the preceding heat and sentiments of the movie dimensions into the ARO model.

## 4. Results

In this section, we compare the ARHS model with the AR model, the eWOM-based model, the GSI-based model, and the ARO model to validate its effectiveness.

In this paper, we use the mean absolute percentage error (MAPE) to measure the performance of the predictive models:

$$MAPE = \frac{1}{n} \sum_{i=1}^{n} \frac{|Pred_i - True_i|}{True_i} \times 100\% \tag{7}$$

where $n$ is the number of predictions made on the test data, $Pred_i$ is the predicted box-office revenues, and $True_i$ represents the true value of the box-office revenues. In statistics, *MAPE* is a suitable measure of accuracy for time-series-value predictions. We can compare the error of the fitted time series because it is a percentage error. All the *MAPE* results reported herein are the mean value of the independent runs of 122 movies on different days. This metric is robust to comparing the performance of the sales prediction models [5,57]. For brevity, we removed the percent sign (%) of the *MAPE* value from Figures 6–9.

### 4.1. Performance of the Parameters in the ARHS Model

In the ARHS model, the parameters $p$, $q$, $\gamma$, and $\delta$ provide the flexibility to fine tune the model for optimal performance. We now explore how the choices of these parameter values affect prediction accuracy.

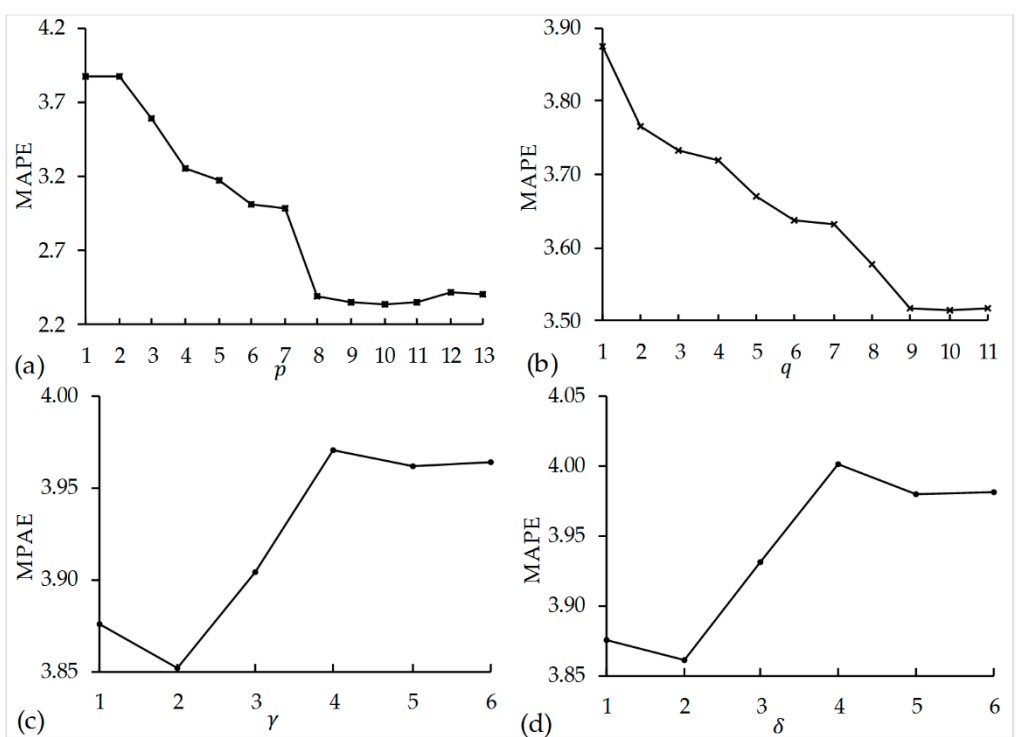

**Figure 6.** The effects of the parameters on prediction accuracy. (**a**) Effects of $p$; (**b**) effects of $q$; (**c**) effects of $\gamma$; (**d**) effects of $\delta$.

First, we varied $p$ with fixed values of parameters $q$, $\gamma$ and $\delta$ ($q = \gamma = \delta = 1$) to study how preceding box-office revenues affect the prediction accuracy of the ARHS model. As shown in Figure 6a, the model achieves its best prediction accuracy when $p = 10$. The change in accuracy is minor after $p = 8$, and the accuracy even decreases after $p = 11$. These findings suggest that $p$ should be large enough to factor in all significant influences of preceding box-office revenues but that it should not be so large that it lets irrelevant preceding box-office revenues reduce prediction accuracy.

Next, using a fixed value of $p$, $\gamma$, and $\delta$ ($p = \gamma = \delta = 1$), we varied the value of $q$ from 1 to 11 to study its effect on prediction accuracy. Figure 6b shows that the model achieves its best performance when $q = 10$. However, the accuracy is basically the same after $q = 9$, which means that numerical

online information will affect box-office revenues over the following nine days. Based on the above results, we suggest that the predictive power of numerical online information for box-office revenues lasts slightly longer than the preceding box-office revenues.

By using fixed values for $p$, $q$, and $\delta$ ($p = q = \delta = 1$), we varied $\gamma$ from 1 to 6 to study the prediction accuracy of the ARHS model. As shown in Figure 6c, the ARHS model achieves the best prediction accuracy at $\gamma = 2$, which implies that the effect of the heat of dimensions captured from the text of eWOM lasts two days.

We also varied $\delta$ from 1 to 6, using fixed values for $p$, $q$, and $\gamma$ ($p = q = \gamma = 1$). As shown in Figure 6d, the ARHS model achieves the highest accuracy at $\delta = 2$, which implies that the effects of the sentiments of dimensions on box-office revenues also last two days.

From the results above, we conclude that the product-dimension information captured from online comments has a shorter effect on box-office revenues than numerical online information. We think the reason for this result is that consumers look through the text of eWOM posted only in recent days but glance at the numerical information of eWOM posted over a longer period of time before they decide to see a movie. The optimal parameter values of the ARHS model should be simultaneously searched ($p$, $q$ from 1 to 12; $\delta$, $\gamma$ from 1 to 6). After comparing 4356 experimental results, we found that the optimal parameter values are $p = 8$, $q = 9$, and $\delta = \gamma = 2$.

### 4.2. Comparison of the Predictive Models

To verify the superiority of the ARHS model, we compared its performance with that of the other models.

First, we compared the ARHS model ($q = 9$, $\delta = \gamma = 2$) with the AR model. As shown in Figure 7, the ARHS model consistently outperforms the AR model as $p$ ranges from 1 to 10. We observe that the ARHS model has much higher accuracy when $p$ is small, which implies that the eWOM of a movie has more predictive power when we know little about the preceding box-office revenues. When $p = 4$, our proposed sales prediction model improves the MAPE of the AR model by 27.65%. When the lag of sales is 8, the improvement of the MAPE is the smallest, 2.69%. These improvements suggest that the ARHS model has higher accuracy.

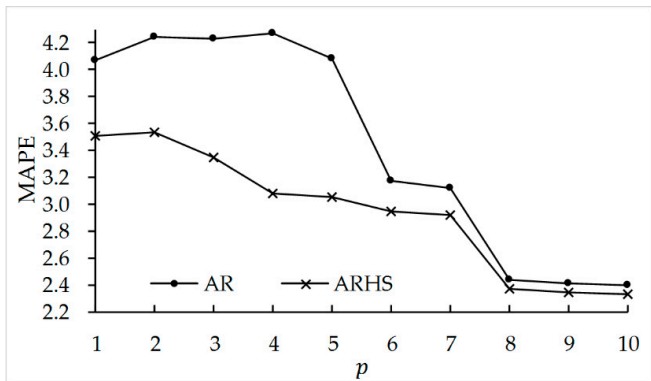

**Figure 7.** Comparison with the autoregressive prediction model.

We then conducted experiments to compare the ARHS ($\delta = q = \gamma = 1$) model with the eWOM model, the GSI model [41], and the ARO model. Both our study and previous studies prove that these models are better than the AR model. As shown in Figure 8a, the eWOM model and the GSI model have nearly the same accuracy performance with regard to forecasting the sales of experience products. As shown in Figure 8b–d, the ARHS model always outperforms the eWOM model, the GSI model, and the ARO model when $p$ ranges from 1 to 6. Thus, the ARHS model is the best among these models, which supports our hypothesis. The effects of the eWOM text on box-office revenues decrease over time, and our test occurs at the end of the release period of the movie. Therefore, compared with the

eWOM model, the GSI model, and the ARO model, the ARHS model improves the *MAPE*, but not much. We argue that the improvement in accuracy of the ARHS model will be higher earlier after a movie is released. Because of the high gross of movies, a very small improvement in forecasting accuracy might result in a difference of millions of dollars. Therefore, the ARHS model should be meaningful to movie marketers and theater managers.

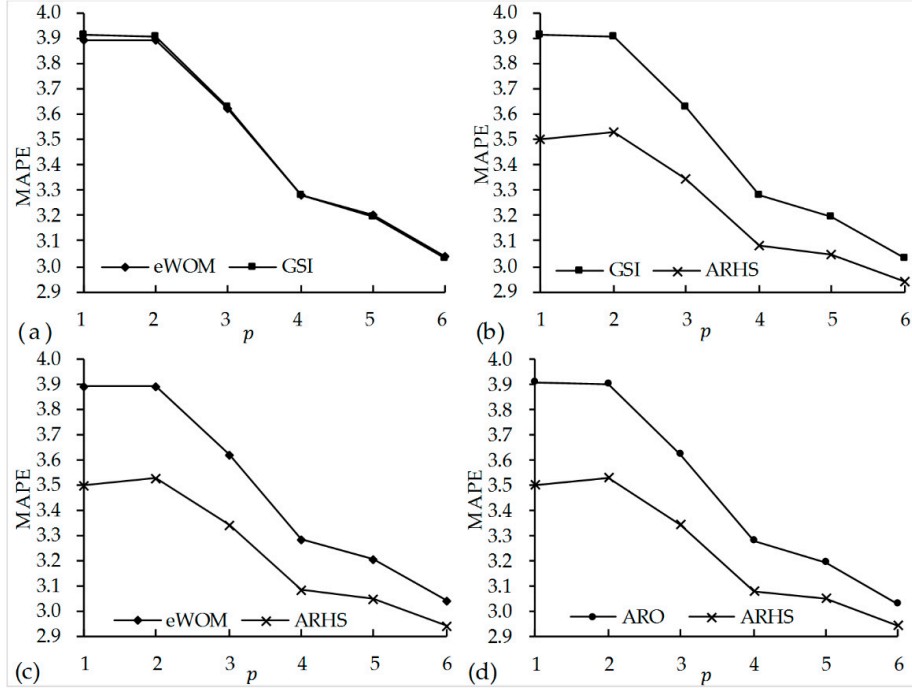

**Figure 8.** Comparisons of model accuracy. (**a**) Comparison of the eWOM model and the GSI model; (**b**) comparison of the GSI model and the ARHS model; (**c**) comparison of the eWOM model and the ARHS model; (**d**) comparison of the ARO model and the ARHS model.

To verify the time robustness of the ARHS model, we compared its accuracy for different predictive periods. We use the first 20, 30, and 40 days of data as training data and the following 9 days of data as test data. Figure 9 shows the results. The prediction accuracy increases when $p$ $(0 < p < 8)$ increases, and it barely changes after $p \geq 8$. The prediction accuracy for 21–29 days is always higher than that for 31–39 days, and the prediction accuracy for 31–39 days is higher than that for 41–49 days. This means that the prediction performance of the ARHS model is higher in the initial stage of a movie's release and that earlier is better. Therefore, we conclude that the heat and sentiments of dimensions have greater predictive power in the early days after a movie's release.

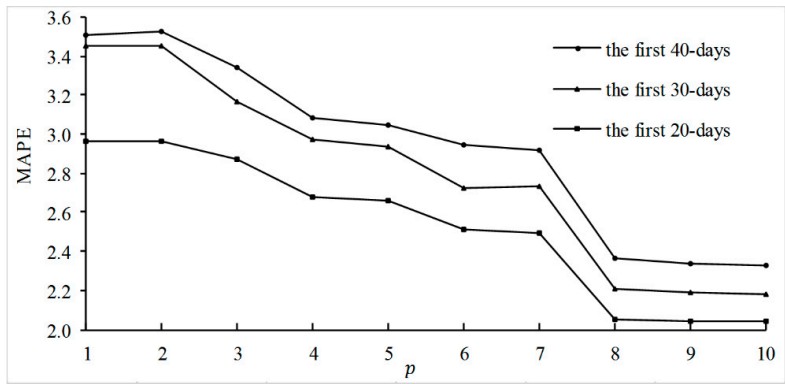

**Figure 9.** Comparison of different prediction intervals.

### 4.3. Robustness of the Predictive Power of the Heat and Sentiments of Dimensions

To further verify the predictive power of the heat and sentiments of dimensions, we conduct a robustness check with regard to predicting the opening-week revenues of movies, which determines the gross of movies, by using a BP neural network with 10-fold cross validation. We filtered out the movies that did not have any online reviews before being released. The final dataset comprises 111 movies with 14,328 online reviews, Google Trends and film-related factors; however, it does not include preceding revenues.

We calculated the overall sentiment of the reviews using the previous method [12]. To verify the forecasting performance of the extracted dimension-specific information, we included the predicted factors in the ARO model without volume in the BP neural network as a benchmark model [41]. We then constructed a new predictive model by integrating the volume and overall sentiment into the benchmark model, which we call the volume–overall–sentiment (VOS) model. Finally, we constructed the dimension-heat-sentiment (DHS) model by integrating the heat and sentiments of dimensions. We used the root mean squared error (*RMSE*) and *MAPE* to measure the prediction accuracy of the BP neural network:

$$RMSE = \sqrt{\frac{1}{n} \sum_{n=1}^{n} (y_i - \hat{y}_i)^2} \tag{8}$$

where $n$ is the sample size, $y_i$ is the actual box-office revenues, and $\hat{y}_i$ represents the predicted box-office revenues.

Figure 10 shows the *RMSE* and *MAPE* of these models for predicting the opening-week box-office revenues. We found that the VOS model is always better than the benchmark model and that the DHS model is better than the VOS for opening-week revenue prediction. Therefore, the heat and sentiments of dimensions have better prediction performance for movie opening-week revenues when the machine-learning method is used.

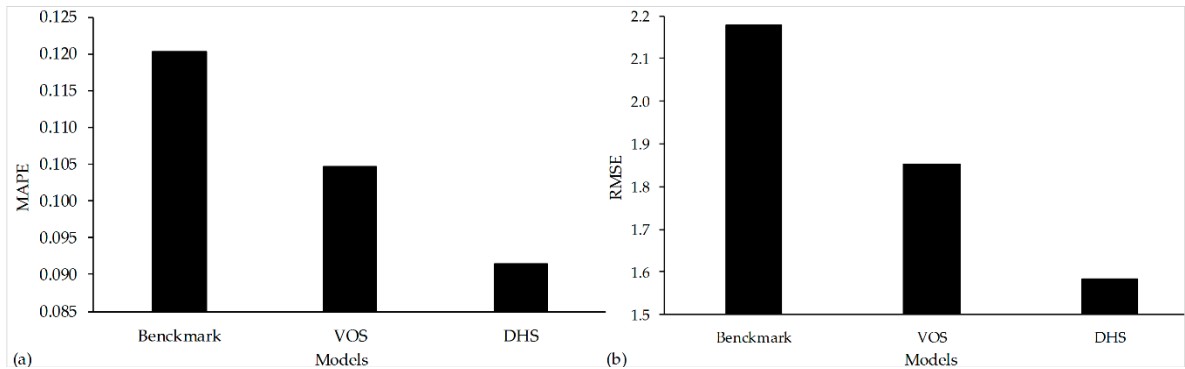

**Figure 10.** Accuracy of models predicting opening-week revenues. (**a**) *MAPE* of the three models; (**b**) *RMSE* of the three models.

### 5. Conclusion and Discussion

Previous research demonstrates that eWOM text implies the heat and sentiments of product dimensions that influence product sales [11,12,43]. Thus, we propose a method called DTA to extract the heat and sentiments of product dimensions from big data on eWOM. Previous research has proven that the multiattribute attitude model with three attributes is sufficiently concise and effective [43,58]. Based on our DTA results, we obtained the dynamic heat and sentiments of three key movie dimensions: the *plot*, the *genre*, and the *star*. These dimensions have also been regarded as important movie dimensions in previous studies, but the heat and sentiments of these dimensions had not been investigated [7,53]. To improve accuracy, we propose the ARHS model by integrating the heat and sentiments of dimensions into a prediction model for movie daily ticket sales. This model's performance was compared with that of other predictive models, and the results indicate that the

ARHS model is more accurate than the benchmark model [5], which supports our hypothesis. We also found that the ARHS model performs much better in the early stage of product release. The robustness check with regard to predicting opening-week revenues using the machine-learning method also demonstrates that the heat and sentiments of dimensions have more predictive power.

Our research has some theoretical implications. First, our research extends the use of multiattribute attitude theory to a big data environment using DTA, which is a framework based on a machine-learning method, syntactic method, and lexicon-based method. In previous research, attributes used to predict consumer purchase predispositions were generated based on expert judgment and in-depth interviews. Additionally, the measures of attribute importance and valence were obtained by surveying a sample of respondents [43]. However, individual deviations, the limited number of survey samples and halo effects may bias the results [59], and these methods often have high manpower and time costs that are not suitable for big data. Therefore, we propose DTA to extract the heat and sentiments of the most important dimensions from eWOM big data as a proxy for attribute importance and valence in social media marketing research, making it possible to avoid the issues of the methods used in previous studies. Second, we demonstrate that multiattribute attitude theory [9,10] is valid for forecasting sales in a big data environment. Previous research used eWOM volume and sentiment as a whole to predict sales, with limited performance. Our research shows that the heat and sentiment of dimensions extracted from social media based on multiattribute attitude theory can improve the predictive performance of the benchmark model. Third, compared with the results regarding search products [5], our study proves that integrating social media data and online search data together can improve the performance of sales predictions for experience products. Fourth, our research demonstrates that the heat and sentiments of dimensions implied in social media improve the accuracy of predictive models. Therefore, related research, such as studies that predict election outcomes, stock prices, and internet security, can use the heat and sentiments of dimensions to construct predictive models.

Additionally, our paper has some practical implications. First, using the DTA, we can extract the heat and sentiments of dimensions in a timely way and can then monitor consumer feedbacks on brand dimensions by analyzing the dynamic heat and sentiments of brand dimensions over time. There may be an initial collaborative attack on a brand dimension when the heat of the brand dimension suddenly increases and the sentiment of the brand dimension decreases suddenly and simultaneously. At that point, we can provide a warning of "collaborative brand attacks" to enterprises. The stakeholders can then confirm it in social media and carry out certain activities to stop the attack if the information is accurate. Second, we find that the predictive power of the volume and ratings of eWOM lasts longer than that of eWOM text: the heat and sentiments of product dimensions. Therefore, managers should pay attention to the volume and ratings of eWOM over a long period and pay attention to the text of new eWOM only. Third, the proposed ARHS model has better predictive performance for sales. Therefore, marketing managers can provide an early warning with regard to a sales explosion or collapse and more accurately determine whether and when to carry out promotion activities using our method. In the movie industry, marketers can provide an early warning with regard to ticket sales before a movie is released, and theaters can adjust the number of screens for different movies based on accurate daily box-office predictions. Fourth, we can determine the strength and weakness of relevant product attributes by comparing the DTA results of different products. For example, if one dimension of a product has the lowest sentiment among all competitive products, this dimension can be the weakness of the product. Therefore, this method can also be used to suggest specific changes in product design and the marketing support for the product.

This paper also has some limitations. Our research can only benefit the social media marketing of products and brands with multiple attributes and abundant social media data. We predict only box-office revenues in America to demonstrate the predictive power of the heat and sentiments of dimensions. To improve our theory, we plan to conduct further research that forecasts the sales of other products in different regions. Additionally, in this paper, we use only one type of eWOM and

social media data. We should use multiple types of eWOM and data in other forms of media in future research to obtain results that are more robust. Future research can be conducted to examine how dimension-specific information influences product sales differently, which can help to identify the different economic effects of each product attribute.

**Author Contributions:** Conceptualization, X.L. and C.J.; methodology, X.L.; software, X.L.; validation, X.L., Y.D., and Z.W.; formal analysis, X.L. and Y.L.; investigation, X.L.; resources, X.L.; data curation, X.L.; writing—original draft preparation, X.L.; writing—review and editing, X.L. and C.J.; visualization, X.L.; supervision, X.L., Z.W., and Y.L.; project administration, C.J. and Y.D.; funding acquisition, C.J.

**Funding:** This research was funded by the National Natural Science Foundation of China (NSFC), key grant number 71731005 and grant number 71571059.

**Conflicts of Interest:** The authors declare no conflict of interest. The funders had no role in the design of the study; in the collection, analyses, or interpretation of data; in the writing of the manuscript; or in the decision to publish the results.

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
