# Peer review of "Sales Prediction by Integrating the Heat and Sentiments of Product Dimensions"

_sustainability, doi:10.3390/su11030913_

Round 1
Reviewer 1 Report
This topic is contemporary and of interest for researchers and
practitioners, while your data and method of analysis are also creative
and a good choice for this subject.
What you need to improve
is the theoretical base of your paper. First, please clarify your
theoretical framework and provide theoretical support, as well as
hypotheses. Second, your findings should be related to previous
literature and should also emphasize your contributions to research and
practice.
Author Response
Dear reviewer, Thank you for your kindly and professional suggestions. We have revised our paper based on your suggestions, and all revisions are highlighted in the paper. We response to you based on your comments. First, please clarify your theoretical framework and provide theoretical support, as well as hypotheses. Response: We have clarified our research framework and provided theoretical supports (multi-attribute attitude theory, topic model, etc.), as well as hypothesis in Section 3.1. Additionally, we add some new paper as references. The details of revision are as followed: “3.1. Research Framework Figure 1 shows the framework of our study; it will help researchers develop a sales prediction model for products with abundant eWOM and multiple attributes. First, based on the findings that eWOM volume [25,27–29] and ratings [27,36] positively affect sales, we develop the eWOM model by integrating eWOM variables into the autoregressive (AR) model. Additionally, we develop the GSI model by integrating Google Trends into the AR model based on the findings in the literature [41]. Then, we integrate Google Trends into the eWOM model following the method of [5] and name this benchmark model the autoregressive online (ARO) model. The above models consider volume (heat) and valence as a whole. Multiattribute attitude theory decomposes a consumer’s overall attitude toward a product into smaller components [9,10]. These components, which are the most important attributes, reflect only consumers’ perceptual dimensions rather than product characteristics that are directly controllable and measurable by marketing managers [33]. Multiattribute attitude theory shows that only the importance and valence of important product attributes can predict consumer purchase predispositions. Volume and sentiment, which represent the sum of the heat and sentiments of all attributes implied in eWOM, may have little effect on sales because of the offset effects of irrelevant and redundant attributes. Perhaps for this reason, some research finds that the volume and valence of eWOM have no effect on sales [15,38,39]. In previous studies, attributes were generated based on expert judgment and in-depth interviews. Additionally, the measures of attribute importance and valence in the attitude model were obtained by surveying a sample of respondents [43]. Recent research shows that topic models can be used to generate important attributes that reflect consumers’ perceptual dimensions, and the heat and sentiment of these product dimensions can be used as a proxy for attribute importance and valence [44–46]. Therefore, the heat and sentiments of dimensions are more suitable than eWOM volume and valence for predicting sales. Thus, according to the discussion above, we hypothesize the following: H1. The heat and sentiments of dimensions have predictive power for sales and can improve the performance of the benchmark model. Finally, to verify the hypothesis, we use DTA to extract the heat and sentiments of product dimensions from eWOM and integrate them into the benchmark model to determine whether the new model, the ARHS model, has better prediction accuracy. Figure 1. Framework for constructing the ARHS model. ” Second, your findings should be related to previous literature and should also emphasize your contributions to research and practice. Response: We have related our findings to previous literatures in the first paragraph of Section 5. The details of revision are as followed: “Previous research demonstrates that eWOM text implies the heat and sentiments of product dimensions that influence product sales [11,12,43]. Thus, we propose a method called DTA to extract the heat and sentiments of product dimensions from big data on eWOM. Previous research has proven that the multiattribute attitude model with three attributes are sufficiently concise and effective [43,58]. From our DTA results, we obtain the dynamic heat and sentiments of three key movie dimensions: the plot, the genre and the star. These dimensions were also regarded as important movie dimensions in previous studies, but they did not investigate the heat and sentiments of these dimensions [7,53]. Then, to improve accuracy, we propose the ARHS model by integrating the heat and sentiments of dimensions into a prediction model for movie daily ticket sales. This model’s performance is compared with that of other predictive models, and the results indicate that the ARHS model is more accurate than the benchmark model [5] that support our hypothesis. We also find that the ARHS model performs much better in the early stage of product release. The robustness check with regard to predicting opening-week revenues using the machine-learning method also demonstrates that the heat and sentiments of dimensions have more predictive power. ” Our contributions to research and practice are emphasized in the second and third paragraph. The details of revision are as followed: “Our research has some theoretical implications. First, our research extends the use of multiattribute attitude theory to a big data environment using DTA, which is a framework based on the machine-learning method, syntactic method and lexicon-based method. In previous research, attributes used to predict consumer purchase predispositions were generated based on expert judgment and in-depth interviews. Additionally, the measures of attribute importance and valence were obtained by surveying a sample of respondents [43]. However, individual deviations, the limited number of survey samples and halo effects may bias the results [59], and these methods often have high manpower and time costs, which are not suitable for big data. Therefore, we propose DTA to extract the heat and sentiments of the most important dimensions from eWOM big data as a proxy for attribute importance and valence in social media marketing research, making it possible to avoid the issues of the methods used in previous studies. Second, we demonstrate that multiattribute attitude theory [9,10] is valid for forecasting sales in a big data environment. Previous research uses eWOM volume and sentiment as a whole to predict sales, having limited performance. Our research shows that the heat and sentiment of dimensions extracted from social media based on multiattribute attitude theory can improve the predictive performance of the benchmark model. Third, compared with the results regarding search products [5], our study proves that integrating social media data and online search data together can also improve the performance of sales predictions for experience products. Fourth, our research demonstrates that the heat and sentiments of dimensions implied in social media do indeed improve the accuracy of predictive models. Therefore, related research, such as studies that predict election outcomes, stock prices and internet security, can use the heat and sentiments of dimensions to construct predictive models. Additionally, our paper has some practical implications. First, using the DTA, we can extract the heat and sentiments of dimensions timely and then can monitor consumer feedbacks on brand dimensions by analyzing the dynamic heat and sentiments of brand dimensions over time. There may be a beginning of collaborative attack on a brand dimension when the heat of the brand dimension increases suddenly and the sentiment of the brand dimension decreases suddenly and simultaneously. At that point, we can provide a warning of “collaborative brand attacks” to enterprises. Then, the stakeholders can confirm it in social media and carry out certain activities to stop the attack if the attack is true. Second, our research find that the predictive power of the volume and ratings of eWOM lasts longer than that of eWOM text: the heat and sentiments of product dimensions. Therefore, managers should pay attention to the volume and ratings of eWOM over a long period and pay attention to the text of new eWOM only. Third, the proposed ARHS model has better predictive performance for sales. Therefore, marketing managers can provide an early warning with regard to a sales explosion or collapse and determine whether and when to carry out promotion activities more accurately using our method. In the movie industry, marketers can provide an early warning with regard to ticket sales before a movie is released, and theaters can adjust the number of screens for different movies based on accurate daily box-office predictions. Fourth, we can determine the strength and weakness of relevant product attributes by comparing the DTA results of different products. For example, if one dimension of a product has the lowest sentiment among all competitive products, this dimension can be the weakness of the product. Therefore, this method can also be used to suggest specific changes in product design and the marketing support for the product. ” Thank you again for your kindly and constructive comments. We look forward to hearing from you regarding our submission. We would be glad to response to any further questions and comments that you may have.

Reviewer 2 Report
This is an interesting research on WOM and its relationship with sales.
I do have, however, few comments:
First, I would like to see a broader discussion on how this study contributes to social media marketing as a whole. Therefore, I strongly recommend the following changes: Start with a broad definition of social media, such as in Felix et al (2016), Elements of Strategic Social Media Marketing. These definitions encompass monitoring as an important aspect, so I would like to see this in the frontend of the paper, as well as in the discussion at the end. In addition, at the end of the paper, I would like to see discussions how your findings, in a practical context, can be used more extensively. More specifically: What can companies do with that? How about early warning systems for "Collaborative Brand Attacks"? Or Sales forecasts? Be a bit more philosophical, such as: How can you link sentiment with weather data and so forth?
Second, the methods part is very "maths"focused, which papers in this journal are typically not. Please add some more verbald descriptions and examples around the formulas. Less "maths-focused" readers will be thankful for that. This also includes the abstract.
Third, future research could also look at other forms of media.
Please justify the methods better. Why autoregressive and not ai tools?
Please use copy editing
Author Response
Dear reviewer,
Thank you for your constructive and professional suggestions. We have revised our paper based on your suggestions, and all revisions are highlighted in the paper. We response to you based on your comments.
First, I would like to see a broader discussion on how this study contributes to social media marketing as a whole. Therefore, I strongly recommend the following changes: Start with a broad definition of social media, such as in Felix et al (2016), Elements of Strategic Social Media Marketing. These definitions encompass monitoring as an important aspect, so I would like to see this in the frontend of the paper, as well as in the discussion at the end. In addition, at the end of the paper, I would like to see discussions how your findings, in a practical context, can be used more extensively. More specifically: What can companies do with that? How about early warning systems for "Collaborative Brand Attacks"? Or Sales forecasts? Be a bit more philosophical, such as: How can you link sentiment with weather data and so forth?
Response:
Thank you for your kindly suggestions. We start our introduction with the definition of social media and discuss more on social media marketing, and we have read and referenced the paper “Felix, R.; Rauschnabel, P.A.; Hinsch, C. Elements of strategic social media marketing: A holistic framework. J. Bus. Res. 2017, 70, 118–126.”. The details of revision are as followed:
“Social media are forms of electronic communication (such as Facebook, WeChat and IMDb.com) through which people create online communities to share information, ideas and personal messages. Increasingly more consumers use the online word-of-mouth (eWOM) on social media for decision support before making purchases [1]. Therefore, social media marketing has recently appeared as an interdisciplinary and cross-functional concept that uses social media to achieve organizational goals by creating value for stakeholders [2]. Sales prediction is a foundation for social media marketing. Highly accurate and timely sales predictions can allow firms to reduce their profit losses and to improve their market performance [3]. Due to the superiority of big data on social media, sales predictions are being produced more than ever before to increase their accuracy and to enable them to support real-time marketing strategies for online retailers and enterprises. However, the accuracy of these models is still unsatisfactory. We need to extract more predictive information from high-frequency social media data to improve the accuracy of sales predictions.”
We have explained our contributions more on social media marketing in Section 5. The sentiment may be used to predict the election, stock price and internet security. Unfortunately, we cannot link sentiment with weather data with the best of our knowledge. We are not familiar with weather data so that we cannot link sentiment with weather data. The details of revision are as followed:
“Our research has some theoretical implications. First, our research extends the use of multiattribute attitude theory to a big data environment using DTA, which is a framework based on the machine-learning method, syntactic method and lexicon-based method. In previous research, attributes used to predict consumer purchase predispositions were generated based on expert judgment and in-depth interviews. Additionally, the measures of attribute importance and valence were obtained by surveying a sample of respondents [43]. However, individual deviations, the limited number of survey samples and halo effects may bias the results [59], and these methods often have high manpower and time costs, which are not suitable for big data. Therefore, we propose DTA to extract the heat and sentiments of the most important dimensions from eWOM big data as a proxy for attribute importance and valence in social media marketing research, making it possible to avoid the issues of the methods used in previous studies. Second, we demonstrate that multiattribute attitude theory [9,10] is valid for forecasting sales in a big data environment. Previous research uses eWOM volume and sentiment as a whole to predict sales, having limited performance. Our research shows that the heat and sentiment of dimensions extracted from social media based on multiattribute attitude theory can improve the predictive performance of the benchmark model. Third, compared with the results regarding search products [5], our study proves that integrating social media data and online search data together can also improve the performance of sales predictions for experience products. Fourth, our research demonstrates that the heat and sentiments of dimensions implied in social media do indeed improve the accuracy of predictive models. Therefore, related research, such as studies that predict election outcomes, stock prices and internet security, can use the heat and sentiments of dimensions to construct predictive models.
Additionally, our paper has some practical implications. First, using the DTA, we can extract the heat and sentiments of dimensions timely and then can monitor consumer feedbacks on brand dimensions by analyzing the dynamic heat and sentiments of brand dimensions over time. There may be a beginning of collaborative attack on a brand dimension when the heat of the brand dimension increases suddenly and the sentiment of the brand dimension decreases suddenly and simultaneously. At that point, we can provide a warning of “collaborative brand attacks” to enterprises. Then, the stakeholders can confirm it in social media and carry out certain activities to stop the attack if the attack is true. Second, our research find that the predictive power of the volume and ratings of eWOM lasts longer than that of eWOM text: the heat and sentiments of product dimensions. Therefore, managers should pay attention to the volume and ratings of eWOM over a long period and pay attention to the text of new eWOM only. Third, the proposed ARHS model has better predictive performance for sales. Therefore, marketing managers can provide an early warning with regard to a sales explosion or collapse and determine whether and when to carry out promotion activities more accurately using our method. In the movie industry, marketers can provide an early warning with regard to ticket sales before a movie is released, and theaters can adjust the number of screens for different movies based on accurate daily box-office predictions. Fourth, we can determine the strength and weakness of relevant product attributes by comparing the DTA results of different products. For example, if one dimension of a product has the lowest sentiment among all competitive products, this dimension can be the weakness of the product. Therefore, this method can also be used to suggest specific changes in product design and the marketing support for the product.”
Second, the methods part is very "maths" focused, which papers in this journal are typically not. Please add some more verbal descriptions and examples around the formulas. Less "maths-focused" readers will be thankful for that. This also includes the abstract.
Response:
Thank you for your kindly suggestions. We have add some verbal descriptions after the formulas and in the abstract. The details of revision are as followed:
Line 290-291
“The AR model uses only preceding sales to predict current or future sales.”
Line 201-202
“The ARO model uses preceding sales, Google Trends, the eWOM variables and other predictors in Table 4 to predict current and future sales.”
Line 313-314
“The ARHS model extends the ARO model by integrating the preceding heat and sentiments of the movie dimensions into the ARO model.”
the abstract
“Online word-of-mouth (eWOM) disseminated on social media contains a considerable amount of important information that can predict sales. However, the accuracy of sales prediction models using big data on eWOM is still unsatisfactory. We argue that eWOM contains the heat and sentiments of product dimensions, which can improve the accuracy of prediction models based on multiattribute attitude theory. In this paper, we propose a dynamic topic analysis (DTA) framework to extract the heat and sentiments of product dimensions from big data on eWOM. Ultimately, we propose an autoregressive heat-sentiment (ARHS) model that integrates the heat and sentiments of dimensions into the benchmark predictive model to forecast daily sales. We conduct an empirical study on the movie industry confirming that the ARHS model is better than other models in predicting movie box-office revenues. The robustness check with regard to predicting opening-week revenues based on a back-propagation neural network also suggests that the heat and sentiments of dimensions can improve the accuracy of sales predictions when the machine-learning method is used.”
Third, future research could also look at other forms of media.
Response:
Thank you for your remind. We have added this in the future research in Section 5. The details of revision are as followed:
Line 484-486
“Additionally, in this paper, we use only one type of eWOM and social media data. We should use multiple types of eWOM and data in other forms of media in future research to obtain results that are more robust.”
Please justify the methods better. Why autoregressive and not ai tools?
Response:
Now, we have explained why we used autoregressive model and not machine learning models in the first paragraph of Section 3.3. The details of revision as followed:
Line 273-280
“We use the first 40-day data to train the predictive model and the last 9-day data to test the trained model. The regressive model can have better forecasting performance than the machine-learning models when the amount of relevant information is sufficient, and when the variation in box-office revenues is small [55]. However, if there is not enough information, then the machine-learning model can help improve the forecasting accuracy by utilizing the limited information given more thoroughly. According to the sufficient predictors discussed in Section 3.2 and the relatively stable revenues in the test period, the proposed approach that we construct is based on the autoregressive model because the regressive model is the most efficient predictive model [7].”
Please use copy editing
Response:
Thank you for your suggestion. We have used the America Journal Experts (AJE) to do the copy editing for this paper.
Thank you again for your kindly and constructive comments. We look forward to hearing from you regarding our submission. We would be glad to response to any further questions and comments that you may have.

This manuscript is a resubmission of an earlier submission. The following is a list of the peer review reports and author responses from that submission.
Round 1
Reviewer 1 Report
This paper proposes a new framework—dynamic topic analysis (DTA) framework—to predict the sales of a product. The framework is based on the premise that the predictive accuracy of a model can be improved by combining the heat (volume) and sentiment of product dimensions in user reviews with online search data. The authors use the proposed framework to predict movie ticket sales. They collect daily reviews, ticket sales, online search index of 122 movies for the first 50 days of release. They train their proposed model with the first 40 days of data and use the trained model to predict the ticket sales of the following nine days. They compare the performance of their proposed model with that of benchmark models to show that their model outperforms the competing models. This paper has numerous issues.
1. The originality of the paper
a. The basic premise of the proposed model is that one can improve the predictive performance of a model by combining social media data and online search data. However, this is not new as Geva et al. (2017) have shown.
b. It is also well-known that review volume and sentiment can predict product sales.
c. That said, the original contribution of the paper is using the review volume and sentiment at the product dimension level. In my opinion, this contribution is not big enough.
2. Application of the model
a. Anyone who has studied the motion picture industry will know that the most important prediction task is the prediction of the opening-week revenue. This is because, for most movies, the majority of ticket sales is realized in the first few weeks after release. It makes more sense to apply their proposed model to predict the opening week revenue. But again, their model will not work well for predicting the opening-week revenue as there is little WOM on a movie before it is released.
b. Google search index from Google Trends is a weekly index, while all the other time-series variables in your data set are daily. How did you derive daily search index from the original weekly index?
c. The authors use only three movie attributes: star, plot, and genre. But there are numerous movie attributes. The authors did not explain why the three attributes are most important for their study.
d. According to Table 6, the minimum volume of daily reviews is zero. For zero-review days, how did the authors estimate the review valence?
e. According to Table 6, the minimum sentiment of daily reviews is zero. However, IMDb’s user rating can have a value between 1 and 10.
f. The authors should use a cross-validation technique (e.g., Leave-One-Out Cross-Validation or k-Fold Cross-Validation) to show the predictive performance.
g. The approach that the author used to search for optimal parameters (lines 280 – 298 on page 10) find suboptimal results. The authors fix all the remaining parameters at arbitrary values to find the optimal value for a focal parameter. For example, to search for the optimal values of p, the authors fix q, gamma, and delta at 1. Instead, the optimal parameter values are searched for simultaneously for all parameters.
3. Presentation: There are a lot of presentation issues. Some of them are
a. The paper needs to be copy-edited. There are a lot of typos and grammatical errors.
b. Be consistent in using mathematical notations. For example, why did you use i (not t) to index time on line 204 of page 6?
c. The paper is unnecessarily complex and long. For example, the paper repeats the DTM that has been originally developed in Blei and Laffetry (2006). Instead, the author should point the readers to the original paper. Present only your contribution.
d. Figure 1 is unnecessarily complex.
e. Explain more about the 122 movies. For example, when and where were they released? How did you select them? Are they wide-release movies?
f. Table 1 and the data part of Table 6 (Sales, v1, v2, v5, and v6) can be combined and presented upfront in the data section.
The presentation issues are minor and can be easily resolved. My major concerns are with the originality of the paper and the application of the proposed model.
Reviewer 2 Report
The topic of your study is very interesting, and the methodology and data as well. You need to improve your literature review and theoretical framework and update the section with the latest studies regarding the relationship between eWOM and sales.
Baker, Andrew M., Naveen Donthu, and V. Kumar (2016), “Investigating How Word-of-Mouth Conversations About Brands Influence Purchase and Retransmission Intentions,” Journal of Marketing Research, 53, 2, 225–39, doi:10.1509/jmr.14.0099.
Kimmel, Allan J. and Philip J. Kitchen (2013), “WOM and Social Media: Presaging Future Directions for Research and Practice,” Journal of Marketing Communications, 20, August 2013, 1–16, doi:10.1080/13527266.2013.797730.
Relling, Marleen, Oliver Schnittka, Henrik Sattler, and Marius Johnen (2016), “Each Can Help or Hurt: Negative and Positive Word of Mouth in Social Network Brand Communities,” International Journal of Research in Marketing, 33, 1, Elsevier B.V., 42–58, doi:10.1016/j.ijresmar.2015.11.001.
Sweeney, Jillian C., Geoffrey N. Soutar, and Tim Mazzarol (2012), “Word of Mouth: Measuring the Power of Individual Messages,” European Journal of Marketing, 46, 1/2, 237–57, doi:10.1108/03090561211189310.
Also, your methodology section should be followed by a stronger overview of your findings and how they fit in the current research framework, as well as your contributions to research and managerial implications.